# CHOPPING FORMERS IS WHAT YOU NEED IN VISION

## ABSTRACT

This work presents a new dynamic and fully-connected layer (DFC) that generalizes existing layers and is free from hard inductive biases. Then, it describes how to factorize the DFC weights efficiently. Using the Einstein notation's convention as framework, we define the DFC as a fully connected layer with the weight tensor created as a function of the input. DFC is the non-linear extension of the most general case of linear layer for neural network, and therefore all major neural network layers, from convolution to self-attention, are particular cases of DFCs. A stack of DFCs interleaved by non-linearities defines a new super-class of neural networks: *Formers*. DFC has four major characteristics: i) it is dynamic, ii) it is spatially adaptive, iii) it has a global receptive field, and iv) it mixes all the available channels' information. In their complete form, DFCs are powerful layers free from hard inductive biases, but their use is limited in practice by their prohibitive computational cost. To overcome this limitation and deploy DFC in real computer vision applications, we propose to use the CP Decomposition, showing that it is possible to factorize the DFC layer into smaller, manageable blocks without losing any representational power. Finally, we propose ChoP'D Former, an architecture making use of a new decomposition of the DFC layer into five sequential operations, each incorporating one characteristic of the original DFC tensor. Chop'D Former leverages dynamic gating and integral image, achieves global spatial reasoning with constant time complexity, and has a receptive field that can adapt depending on the task. Extensive experiments demonstrate that our ChoP'D Former is competitive with state-of-the-art results on three well-known CV benchmarks, namely Large-Scale Classification, Object Detection, and Instance Segmentation, suppressing the need for expensive architecture search and hyperparameter optimization.

## 1 INTRODUCTION

Convolutional Neural Networks (CNNs) have served as the undiscussed cornerstone of Computer Vision (CV) for the past decade thanks to convolutions, which, despite the hard inductive biases of locally connected and shared weights, are able to summarize spatial content very efficiently (Krizhevsky et al., 2017; Simonyan & Zisserman, 2014; He et al., 2016; Howard et al., 2017; Tan & Le, 2019). Nevertheless, in the 2020s, with the availability of more abundant computing resources, the role of convolutions has been challenged by the advent of Transformers (Vaswani et al., 2017; Dosovitskiy et al., 2020) and a new "spatial-mixing" module, called Self-Attention, characterized by lighter inductive biases and high complexity.

The success of Vision Transformers (ViT) has long been attributed to Self-Attention. However, new findings have recently questioned this narrative. For example, d'Ascoli et al. (2021); Wu et al. (2021); Liu et al. (2021b) highlight the importance of convolutional biases in Transformers for CV. Liu et al. (2022); Yu et al. (2022) demonstrate how macro design choices and training procedures alone can be sufficient to achieve competitive performance regardless of the specific spatial module used. Finally, Cordonnier et al. (2019); Han et al. (2021) comment on the close link between convolution and Self-Attention formulations, hence blurring the line between these seemingly orthogonal operators. Here, we take a new step toward bridging the gap between CNNs and Transformers by providing a unifying and intuitive formulation that clarifies spatial modules' role in modern architectures and links existing work together.

First, we use Einstein's tensor notation combined with tensor CP Decomposition to provide a practical yet principled analysis of existing literature. In essence, the principal ingredients in deep learning architectures are multi-dimensional operations that can naturally be written as decomposed tensor expressions. Here, the Einstein notation provides an elegant way to analyze neural network operators by highlighting differences among layers with an intuitive notation that simplifies multi-dimensional matrix algebra (Kolda & Bader, 2009; Panagakis et al., 2021; Hayashi et al., 2019) with no compromises in formal accuracy. Under this lens, we formalize a generalization of existing layers with a new dynamic, spatially adaptive, and fully connected building block for Neural Networks (the DFC) that represents the general – but computationally complex – operation of extracting the complete set of interactions within the input.

Second, we use DFCs to define a super-class of neural networks, which we call Formers, where the dense and heavy DFC operators are used to create hierarchical representations of the input images. Then, to target real-world applications, usually bounded by tight computational budgets, we explore the use of CP Decomposition to decrease Formers' complexity and integrate different inductive biases in their design. In this light, we show that Transformers' architectures can be seen as one of the possible instances of Formers and go a step further by proposing a new ChoP'D Former variant. ChoP'D Former leverages CP Decomposition, dynamic gating, and integral images to "chop" the general but prohibitively complex DFC into a sequence of efficient transformations that have the potential to retain its full representational power. In particular, we identify five specific modules that can model the dynamicity with respect to the input, the adaptivity with respect to the spatial positions, and the long-range interactions via a dynamic receptive field with an overall complexity independent of the number of input tokens.

Finally, this new perspective allows us to justify the empirical success of (Trans)Formers and disentangle the contributions of each of their characteristics. To do so, we programmatically compare different layers and CP-Decomposed architectures on various small-scale and large-scale CV tasks. Our experiments indicate that CP-Decomposed DFC layers can effectively approximate the full DFC at a significantly reduced cost, considerably outperforming its simplified variants. In conclusion, our contributions can be summarized as follows:

- We provide a unifying view on building blocks for neural networks that generalizes and compares existing methods via Einstein's notation and CP Decomposition, with a notation that deals with multi-dimensional tensor expressions without resorting to heavy tensor algebra.
- We show how to use a complete tensor operator that is spatially adaptive, fully connected, and dynamic (DFC) to create general neural networks, which we dub "Formers".
- We connect our formulation to existing architectures by showing how Transformer and its variants can be seen as a stack of CP-Decomposed DFC operands for neural networks.
- We propose ChoP'D Former, a new variant of Former architecture, which is able to approximate the full DFC with a complexity comparable to a convolution with a small kernel, and is able to match, if not improve, SoTA performance on several benchmarks, including large scale classification, object detection, and instance segmentation.

## 2    EINSTEIN NOTATION FOR NEURAL NETWORKS

At their core, neural networks – and deep learning architectures in particular – are commonly built as a sequence of tensor operations (i.e., *building blocks*) interleaved by point-wise non-linearities. Tremendous interest has been dedicated to the form of such building blocks (e.g., "MLP", "Convolution", "Residual Block", "Dense Block", "Deformable Conv", "Attention", "Dynamic-Conv", etc.) as these are the critical components to extract various meaningful information from the input. In this section, we present a general form of a neural network layer and showcase how the Einstein summation convention can be used as an alternative, short-hand, and self-contained way to represent and relate building blocks for neural networks.

### 2.1    BACKGROUND

**Einstein notation.**    In the rest of the paper, we adopt the notation of Laue et al. (2020). Tensors are denoted with uppercase letters and indices to the dimensions of the tensors are denoted in lowercase subscripts. For instance $X_{ijk} \in \mathbb{R}^{I \times J \times K}$ is a three-dimensional tensor of size $I \times J \times K$ with three modes (or dimensions) indexed by $i \in [1, I]$, $j \in [1, J]$, and $k \in [1, K]$. Using the Einstein notation,

any multiplication among tensors can be written as: $C_{s_3} = \sum_{(s_1 \cup s_2) \setminus s_3} A_{s_1} B_{s_2}$ where $s_1$, $s_2$, and $s_3$ are the index sets of the left argument, the right argument, and the result tensor, respectively. The summation is only relevant for inner products and is made explicit by underlining tensor indexes. As a representative example to illustrate our notation, we review a set of common operations among tensors. Given the tensors of order two $Y, X \in \mathbb{R}^{I \times J}$, their Hadamard product can be written as $Z_{ij} = X_{ij} Y_{ij}$ and is equivalent to the algebraic notation $Z = X \odot Y$. Similarly, their matrix-product can be written as $Z_{ij} = X_{i\underline{k}} Y_{\underline{k}j}$ and is equivalent to the algebraic notation $Z = XY^\top$. Given the tensors of order one $Y \in \mathbb{R}^I$ and $X \in \mathbb{R}^J$, their outer product creates a tensor $Z \in \mathbb{R}^{I \times J}$ as $Z_{ij} = X_i Y_j$. It is equivalent to the algebraic expression $Z = Y^\top X$.

**CP Decomposition.** The CP Decomposition also referred to as CANDECOMP/PARAFAC or polyadic decomposition, is used to express the factorization of a multi-dimensional tensor as a linear combination of components with rank one, and thus generalizes the concept of matrix singular value decomposition (SVD) to tensors (Kolda & Bader, 2009). For example, let $X_{ijk} \in \mathbb{R}^{I \times J \times K}$ be a three-dimensional tensor, then we can define the CP Decomposition $X_{ijk} \approx U^1_{i\underline{r}} U^2_{j\underline{r}} U^3_{k\underline{r}}$ as the approximation of the original tensor from a set of three factor matrices $[U^1_{ar}, U^2_{br}, U^3_{cr}]$. The rank of the tensor $X_{ijk}$ is defined as the smallest number of $R$ components needed to generate an equality in the CP Decomposition. Note that we call a CP Decomposition canonical whenever $R$ is equal to the rank of $X_{ijk}$.

## 2.2 THE DYNAMIC FULLY CONNECTED LAYER FOR NEURAL NETWORKS

A *neural network layer* is a function $f$ that takes as input a tensor $X_{mc}$ composed of $m \in [1, M]$ spatial positions (or tokens) with $c \in [1, C]$ features (or channels) and produces as output a tensor $Y_{nd}$ composed of $n \in [1, N]$ tokens with $d \in [1, D]$ channels:

$$Y_{nd} = f(X_{mc}) \tag{1}$$

In the following, we start by considering the special case where $f$ is a linear function before introducing the more general dynamic fully-connected layer.

**Linear Layers.** The most general instantiation of a *linear* neural network layer is the Fully-Connected layer (FC):

$$Y_{nd} = X_{\underline{mc}} W_{\underline{mnc}d}, \tag{2}$$

parametrized by a four-dimensional weight tensor $W_{mncd}$ used to mix all spatial and channels information in the input. The complexity is $\mathcal{O}(M \cdot N \cdot C \cdot D)$ which makes the FC layer computationally expensive, if not prohibitive, in CV tasks. However, complexity can be reduced by using priors such as weight sharing and local processing. Using the Einsum notation, we show in the Appendix that the convolutional layer, its depth-wise and point-wise variants, and the average pooling layer are special cases of the FC layer.

**Dynamic Layers.** A non-linear generalization of the FC layer can be obtained by turning the weight tensor into a function $g$ of the input: $W_{mncd} = g(X_{mc})$. To illustrate that the tensor $W_{mncd}$ is not constant anymore but the result of a dynamic construction mechanism, we now consider a "batch" of input instances $X_{imc}$ created as a stack of $I$ inputs $[X^1_{mc}, X^2_{mc}, ..., X^I_{mc}]$ and the corresponding batch of output instances $Y_{ind}$ both indexed by the new instance dimension $i \in [1, I]$. We call such a layer a Dynamic Fully-Connected layer (DFC):

$$Y_{ind} = X_{i\underline{mc}} W_{i\underline{mnc}d} \qquad \text{with} \quad W_{imncd} = [g(X^1_{mc}), \ldots, g(X^I_{mc})]. \tag{3}$$

As in the FC layer, the DFC generates the output by mixing all spatial and channels information. On top of that, DFC is *spatially adaptive*, i.e. weights are not shared across spatial positions and *dynamic* or *instance adaptive*, i.e. it processes each input differently. We wish to stress the relationship between FC and DFC: i) every instance of FC is also an instance of DFC (i.e. DFC where function $g$ is constant) and ii) there are instances of DFC that are not FC (i.e. DFC where function $g$ is non-constant). Therefore FC is a special case of DFC. Again, a number of well-known neural network layers can be framed as simplified cases of DFC. We show in the Appendix and using the Einstein notation that this is the case for Self-Attention (Vaswani et al., 2017; Wang et al., 2018), Dynamic Convolution (Wu et al., 2019; Hu et al., 2018), Deformable Convolution (Dai et al., 2017; Zhu et al., 2019).

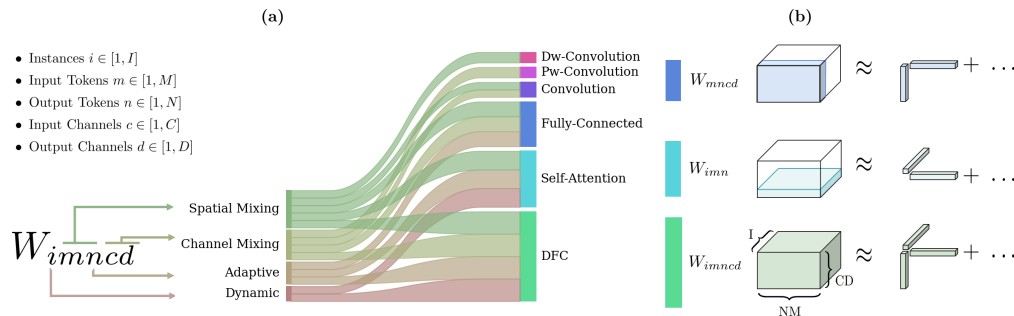

Figure 1: **Part (a): Overview of Building Blocks Characteristics.** The tensor $W_{imncd}$ is a representation of a general neural network layer and each of its dimensions is associated with one characteristic that can be used to describe existing building blocks. For example, a Convolution layer has Token and Channel mixing, but is not Dynamic nor Adaptive. DFC incorporates all possible characteristics in its formulation. **Part (b): Overview of CP Decomposition.** Incorporating additional characteristics in a layer has the side-effect of increasing the dimensions of the underlying parameter tensors. CP decomposition can be used as a tool to approximate the parameter tensor structure while decreasing its complexity. Figure reports examples for a Fully Connected layer (blue), Self-Attention matrix (cyan), DFC layer (green).

As visible in Figure 1, DFC represents a general way to leverage the complete range of interactions of the input and serves as a generalization of existing layers. However, the usage of the DFC is severely limited in practice because of the dense structure of its weights and its high complexity, which is equal to the complexity of the FC layer plus the complexity of the function $g$. Consequently, the question that naturally arises is: *"Is it possible to use these layers efficiently in a neural network without using any strong prior on their weights?"*. Next, instead of relying on sharing and grouping strategies, we propose to use CP Decomposition as a mean to decrease DFC complexity.

## 2.3 "FORMERS" ARCHITECTURES VIA CP DECOMPOSITION

In this section, we describe how to create efficient DFC networks. We use the CP Decomposition as the mathematical backbone to define and design compact and lightweight DFC Neural Networks, in which each layer approximates the behavior of a DFC layer and implements spatial reasoning with a complexity independent of the number of input positions, thus achieving an overall complexity comparable to a standard convolutional layer with a small kernel.

**Formers.** A Former is a general architecture that models hierarchical non-linear interactions by stacking a series of blocks

$$Y_{ind} = \sigma(X_{i\underline{m}c}W_{i\underline{m}n\underline{c}d} + B_{nd}). \tag{4}$$

including a DFC layer with parameters $W_{imncd}$, a matrix of biases $B_{nd}$, and a non-linear element-wise function $\sigma$ (e.g. GeLU or ReLU). In the following, we call DFC layer the linear operation inside the non-linearity, DFC block the DFC layer plus non-linearity, and Former an architecture consisting of one or multiple DFC blocks. As discussed in Section 2.2, the use of a parameters tensor $W_{imncd}$ makes the DFC block general, but also its computation heavy.

**Formers CP Decomposition.** To overcome this limitation, we propose to factorize the weights of the DFC layer through its CP Decomposition:

$$W_{imncd} = U^1_{i\underline{r}}U^2_{m\underline{r}}U^3_{n\underline{r}}U^4_{c\underline{r}}U^5_{d\underline{r}} + \epsilon_{imncd} \tag{5}$$

which represents the full tensor $W_{imncd}$ as a linear combination of lower-dimensional factor matrices plus an approximation error $\epsilon_{imncd}$ dependent on the choice of $R$. Typically, lower $R$ implies larger errors, while for $R \geq rank(W_{imncd})$ the error is zero, and the CP Decomposition is exact.

Hence, we can define the Formers CP Decomposition by replacing the DFC layer in equation 4 with equation 5 and rearranging terms as

$$Y_{ind} = \sigma(((((X_{imn\underline{c}}U^4_{\underline{c}r})_{i\underline{m}nr}U^2_{\underline{m}r})_{inr}U^3_{nr})_{inr}U^1_{ir})_{in\underline{r}}U^5_{d\underline{r}} + B_{nd}) \tag{6}$$

where $X_{imnc}$ is the result of unfolding the input $X_{imc}$ with a global receptive field of size $N$ for all $N$ output positions[1]. Equation 6 acts as a low-complexity substitute of equation 4 and can be

---

[1]In general, we define as "unfolding with receptive field $K$" the operation of rearranging the input as a collection of $N$ sliding patches of size $K$.

easily learned end-to-end as a sequence of fully differentiable building blocks for neural networks. In practice, as apparent from the superscripts of the $U$ matrices in equation 6, we identify five individual – and specific – operations inside a DFC block: i) $U_{cr}^4$: a channel-mixing layer embedding the $C$ input channels into the $R$-space described by the CP Decomposition; ii) $U_{mr}^2$: a spatial layer mixing spatial information with a global receptive field, implemented as depth-wise convolution; iii) $U_{nr}^3$: a gating layer ensuring spatial adaptivity by modulating spatial information; iv) $U_{ir}^1$: a gating layer generating a dynamic response conditioned on the input $X_{imc}$; v) $U_{dr}^5$: a channel-mixing layer which combines the $R$ channels to create $D$ output channels. Replacing DFC layers with the CP Decomposition of equation 6 reduces drastically the memory needed to store the weights from $\mathcal{O}(M \cdot N \cdot C \cdot D)$ to $\mathcal{O}(L \cdot R)$, $L = max(M, N, C, D)$. It also reduces its computational complexity as the sum of its sequence of operations: $\mathcal{O}(M \cdot C \cdot R) + \mathcal{O}(M \cdot N \cdot R) + \mathcal{O}(N \cdot R) + \mathcal{O}(R) + \mathcal{O}(N \cdot D \cdot R)$, plus the complexity of the function used to create matrix $U_{ir}^1$.

Equation 6 is the most general representation of the CP-Decomposed Former, derived directly from applying the CP Decomposition to equation 4. Intriguingly, particular cases of equation 6 can be derived by making assumptions on the factor matrices, thus generating alternative Formers architectures characterized by different inductive biases and complexities. Next, we showcase two particular approximation cases of equation 6: the Transformers, characterized by heavy computational requirements, and the Chop'D Former, our new efficient variant.

**Transformer.** The Transformer (Vaswani et al., 2017) is one of the most well-established and recognized designs for neural networks. The architecture is built from a cascade of inverted residual bottleneck blocks, including a multi-headed self-attention block, two channel-mixing layers, and a GeLU non-linearity[2]. As introduced above, the transformer block is also a particular case of equation 6 as

$$Y_{ind} = \sigma(((X_{imn\underline{c}} \, U_{\underline{c}r}^4)_{i\underline{m}nr} U_{i\underline{m}nr}^{123})_{in\underline{r}} U_{dr}^5 + B_{nd})$$
$$= \sigma((((((X_{imn\underline{c}} \, W_{\underline{c}r}^1)_{imn\underline{r}} W_{\underline{r}r}^2)_{i\underline{m}nr}) U_{i\underline{m}nr}^{123})_{in\underline{r}} V_{d\underline{r}}^2)_{in\underline{r}} V_{dr}^1 + B_{nd}) \qquad (7)$$

where $\sigma$ is a GeLU nonlinearity, the matrices $U_{cr}^4$ and $U_{dr}^5$ are the channel-mixing layers of the inverted bottleneck, and the remaining $U_{ir}^1, U_{mr}^2, U_{nr}^3$ are combined together into a single $U_{imnr}^{123}$, to create a dynamic spatial mixing layer with global receptive field. Moreover, the $U_{imnr}^{123}$ is assumed to be built via a self-attention mechanism only for a subset $H$ of the $R$ channels and then repeated across the $r$ dimension. In other words, equation 7 is a CP Decomposition for a DFC block with three extra assumptions on its factor matrices[3]. Under a similar light, it is easy to recognize how variants of the transformer block can be analogously framed as CP-Decomposed Formers, under a different set of assumptions for the DFC layer factor matrices. This design can process input of various sizes but has two main disadvantages when compared with equation 6. First, it requires higher memory requirements since the parameter tensor $U_{imnr}^{123}$ has to be generated (and also stored in memory) all at once. Second, its computational complexity is: $\mathcal{O}(M \cdot C \cdot R) + \mathcal{O}(M \cdot N \cdot R) + \mathcal{O}(N \cdot D \cdot R)$, plus the complexity of the self-attention mechanism used to create the tensor $U_{imnr}^{123}$. Moreover, as in equation 6 equation, its complexity scales quadratically with the number of tokens used, which can be computationally really expensive even in the case of moderately sized inputs.

**ChoP'D Formers.** From the discussions above, it might seem that equation 6 is a good candidate for an efficient implementation of DFC neural networks. However, in cases where the spatial size is large compared to the number of channels (i.e. $L = M$ or $L = N$), the factor matrices $U_{mr}^2$ and $U_{nr}^3$ of equation 6 act as computation bottlenecks. In fact, implementing the factor matrix $U_{mr}^2$ as a depth-wise convolution with a global receptive field requires a number of parameters directly proportional to the number of spatial positions. Similarly, the gating layer $U_{nr}^3$ has to allocate a parameter for each of the tokens considered. For these reasons, the application of equation 6 is limited to cases where the spatial size of the data is known in advance and small enough to fit in memory. To overcome such limitations, we propose the following modification for equation 6.

First, we replace the operator $U_{mr}^2$ with a more efficient spatial mixing module based on Summed-Area Table (SAT). SAT, also known as an integral image, is a data structure that can be used to perform fast image filtering (Crow, 1984; Viola & Jones, 2004) and enables the computation of

---

[2]Without lack of generality, we omit at this stage the Layernorm (LN) applied before every block and the residual connections, which are used after every block (Wang et al., 2017).

[3](i) $U_{cr}^4 = W_{\underline{c}r}^1 W_{\underline{r}r}^2$; (ii) $U_{dr}^5 = V_{d\underline{r}}^1 V_{\underline{r}r}^2$; (iii) $U_{imnr}^{123} = U_{imnh}^{123} = softmax(Q_{im\underline{c}h} K_{in\underline{c}h})$.

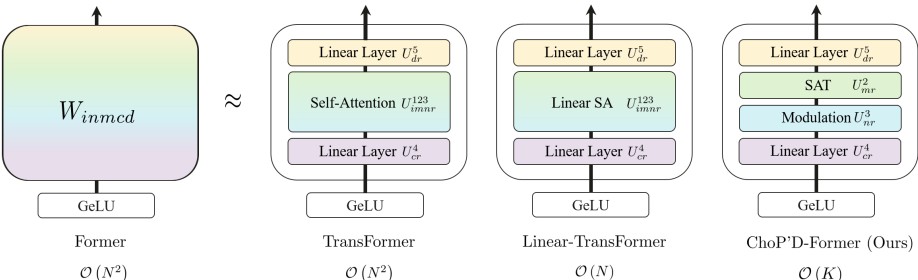

**Figure 2: A Former Architecture** is a stack of DFC layers, in practice often decomposed in a sequence of smaller blocks. Although many alternative decompositions exist for the tensor $W_{imncd}$, they do not come all with the same complexity and inductive bias. The original Transformer has a spatial mixing module that is burdened by a quadratic complexity $\mathcal{O}(N^2)$, unpractical in many real cases. In contrast, our "ChoP'D-Former" token mixer comes with a complexity that is independent of the number of tokens considered.

pooling operations on a receptive field of arbitrary size with a constant computational cost. Additionally, SAT can be used to implement a pooling operation on a *learned* receptive field (Zhang et al., 2019). Thus, we propose to decompose the contribution of the factor matrix $U_{mr}^2$ as follows:

$$U_{mr}^2 = P_{mr\underline{g}}E_{r\underline{g}} \tag{8}$$

where $P_{mrg}$ is a collection of $G$ fully differentiable pooling layers with learnable receptive field, and $E_{rg}$ is the set of learnable weights used to combine their contribution. The advantages are two-fold: i) the model is able to actively learn the optimal receptive field, opting for global or local reasoning for the task at hand, and ii) spatial mixing can be performed at constant computational cost even when the receptive field is global.

Another key modification consists in reducing the memory and computational requirements of the spatial adaptive operators. Specifically, we propose to combine the effect of the two gating $U_{ir}^1$ and $U_{nr}^3$ into a single operator as

$$U_{inr}^{13} = U_{ir}^1 U_{nr}^3 = \phi(X_{imc}), \tag{9}$$

where the function $\phi$, parametrized via a small CNN, generates dynamic and spatially adaptive weights conditioned on the input $X_{imc}$. To further limit complexity, the input of $\phi$ can be downsampled to a pre-defined fixed size, and then the output can be upsampled to match the original resolution, e.g., by interpolation[4]. As a result, the complexity of this spatial adaptive operation is again constant with respect to the number of spatial positions $N$. This allows our formulation to achieve global reasoning with a complexity independent of the number of spatial positions, a drastic improvement when compared with the spatial reasoning module of transformers (i.e. self-attention) and equation 6, both of which have quadratic complexity with respect to the number of tokens. Finally, we can introduce our proposed *CP-Decomposed Formers with learnable Pooling*, or ChoP'D Former for short, by replacing equation 8 and equation 9 in equation 6 as

$$Y_{ind} = \sigma(((((X_{imn\underline{c}}\,U_{\underline{c}r}^4)_{i\underline{m}nr}P_{\underline{m}rg})_{inr\underline{g}})E_{r\underline{g}})_{inr}U_{inr}^{13})_{in\underline{r}}U_{d\underline{r}}^5 + B_{nd}). \tag{10}$$

This formulation, also illustrated in Figure 2, is a CP Decomposition for a DFC block with two extra assumptions on its factor matrices. We recognize several desirable properties: firstly, it can be applied to inputs of arbitrary resolutions without compromises on the size of the receptive field, and secondly, when the number of spatial positions is higher than the number of channels, the overall computational complexity is reduced from $\mathcal{O}(M \cdot N)$ to $\mathcal{O}(M)$.

## 3 RELATED WORK

**Link between Attention and Convolutions.** Han et al. (2021) comments on how the design of local self-attention resembles a dynamic and depth-wise convolution with no weight sharing. Cordonnier et al. (2019) provides proof that a multi-head self-attention layer with a sufficient number of heads is at least as expressive as any convolutional layer. Pérez et al. (2019) shows that transformers are Turing complete. We explore the relationship between these seemingly opposed processing paradigms when interpreted as dynamic layers (Section 2.2, Appendix), and extend this line of research by providing a general framework to compare building blocks as well as architecture designs Section 2.3).

---

[4]The resizing functions are assumed to be absorbed into $\phi$ for the sake of notation simplicity.

**Dynamic Neural Network Layers.** The idea of using a layer whose weights are adaptive to the input can be traced back to early CNNs using max-pooling (Jarrett et al., 2009). Dynamic convolutions emerged multiple times in the context of low-level vision (Jia et al., 2016; Mildenhall et al., 2018; Xia et al., 2020) as well as high-level vision (Ha et al., 2016; Chen et al., 2020; Wu et al., 2019). The dynamic component is also a neglected feature of attention mechanisms (Vaswani et al., 2017; Hu et al., 2018), and we identify it here as the key to unlocking non-linear behavior.

**Tensor Decomposition for Neural Networks.** Tensor Decomposition is an active area of research dedicated to the study of low-rank approximation for multi-dimensional arrays and has applications in a variety of fields, ranging from psychology to CV (Kolda & Bader, 2009; Panagakis et al., 2021). Tensor decomposition techniques have been used to reparameterize neural network layers in order to speed up their inference (Chrysos et al., 2021; 2022; Ma et al., 2019).Lebedev et al. (2014) and Novikov et al. (2015) used CP Decomposition to speed up spatial static convolutional and FC layers. Kossaifi et al. (2020) extended this idea to spatio-temporal static convolutional kernels. Differently from these works, we focus on non-linear dynamic layers and extend this trend of research by investigating a tensor decomposition for a "*Dynamic* Fully Connected" layer (Section 2.3).

**Tensor Notation for Neural Networks.** Einstein notation's convention provides an intuitive notation for tensor manipulations. In machine learning, it can be used as an alternative to tensor algebra (Panagakis et al., 2021; Hayashi et al., 2019). Recently, the Einstein notation has gained traction as a practical way to improve code readability (Rogozhnikov, 2021; Rocktäschel, 2018) and enable efficient tensor calculus (Laue et al., 2020). Here (Section 2.2, 2.3 and Appendix) we use the Einstein notation as a way to compare building blocks for neural networks.

**Summed Area Tables for Neural Networks.** Summed Area Tables (SAT) is an established algorithm in CV (Crow, 1984; Viola & Jones, 2004) that is able to provide the sum of values within an arbitrary subset of a grid in constant time. Recently, SAT has been used to accelerate large-kernel convolution in a dense prediction network for Human Pose Estimation (Zhang et al., 2019) and dynamic large-kernel convolutions in language tasks (Lioutas & Guo, 2020). Similarly in Transformers, SAT enabled fast computation of a linearized attention variant (Zheng et al., 2022) and a parameter-free method to adapt size of the area to attend (Li et al., 2019). As described in Section 2.3, we leverage SAT to achieve an efficient CP Decomposition for a DFC layer and show its application in Formers for CV.

## 4 EXPERIMENTS

In this section, we report the experimental evaluation of Chop'D Former in a wide range of CV tasks. We start by comparing our CP-Decomposed DFC layer of equation 10 against other possible variants in a pre-existing network to assess the contribution of individual components in a controlled setting. Then, we extend our findings to more complex cases by stacking several of such blocks to create architectures with different inductive biases.

**Puzzle Reconstruction** The DFC layer acts as a non-linear extension of an FC layer. It has four main characteristics: being dynamic, being spatially adaptive, having a global receptive field, and mixing all the channel information. To isolate the contribution of each of these characteristics to the overall performance, we compare a DFC layer against its simplified variants: i) a fully connected layer (not dynamic), ii) a convolutional layer (not dynamic, local receptive field), iii) a spatial layer represented by a depth-wise convolution (not dynamic, local receptive field, no channel mixing), iv) a pooling layer (not dynamic, local receptive field, no channel mixing, weights all ones) and v) a channel-mixing layer represented by a point-wise convolution (not dynamic, no spatial mixing). Moreover, we compare its formulation with our efficient CP-Decomposed DFC variant in equation 10 which, we recall, is capable to approximate the full DFC weight tensor $W_{imncd}$ via CP Decomposition. Figure 3 (right) shows the breakdown of different layers in terms of complexity, characteristics, and size of the weight parameter tensors. To compare methods, we use the small-scale but challenging task of "puzzle reconstruction", where a four-layers encoder-decoder network is used to reconstruct an image from a "cut and shuffled" version of itself. Specifically, we obtain input and ground-truth pairs by dividing each sample of the MNIST dataset into 16 different patches, randomly rotating each piece, and shuffling their relative position before stitching them back together. Some examples of input and ground through pairs are visible in Figure 4. We test the ability of different layers to enrich the representation of a network, by placing them between the encoder and the decoder used in an image-to-image translation task. Figure 3 (left) reports validation curves for the compared methods using PSNR as the performance metric, which is a common metric used to assess pixel accuracy (higher is better). Results demonstrate that a DFC layer is able

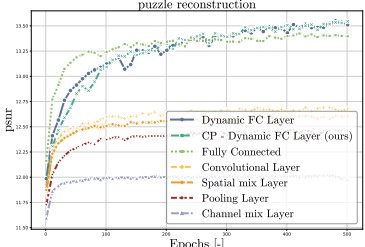

| Layer | Weights Form | $\mathcal{O}()$ | Adaptivity $i, n$ | Receptive Field |
|---|---|---|---|---|
| Dynamic FC | $\mathbf{W_{inmcd}}$ | $\mathcal{O}(N^2 \cdot C^2)$ | $i, n$ | Global |
| Fully Connected | $\mathbf{W_{mncd}}$ | $\mathcal{O}(N^2 \cdot C^2)$ | $n$ | Global |
| Convolutional | $\mathbf{W_{kcd}}$ | $\mathcal{O}(N \cdot K \cdot C^2)$ | - | Local |
| Spatial mixing | $\mathbf{W_{kh}}$ | $\mathcal{O}(N \cdot K \cdot H)$ | - | Local |
| Pooling | $\mathbf{P_{kh}}$ | $\mathcal{O}(N \cdot K \cdot H)$ | - | Local |
| Channel mixing | $\mathbf{W_{cd}}$ | $\mathcal{O}(N \cdot C^2)$ | - | 1 |
| Dynamic FC - CP | $\approx \mathbf{W_{inmcd}}$ | $\mathcal{O}(N \cdot C \cdot R)$ | $i, n$ | Global |

Overview of different Layers

Figure 3: **Puzzle Reconstruction on MNIST. Overview of Layers - right**. Methods are described by complexity and flagged with an $i$ if dynamic, and with an $n$ if spatially-adaptive. $M, N$ indicates input and output tokens, $C, D$ input and output channels, $H$ convolutional groups, $K$ convolutional kernel size, $R$ the CP decomposition size. We assume: $M = N$, $C = D$, $K < N$, $H < C$, $R < C$. **Comparisons among Layers of Neural Networks - left**. PSNR validation curves show how our CP-DFC method is capable to approximate a general DFC layer performance.

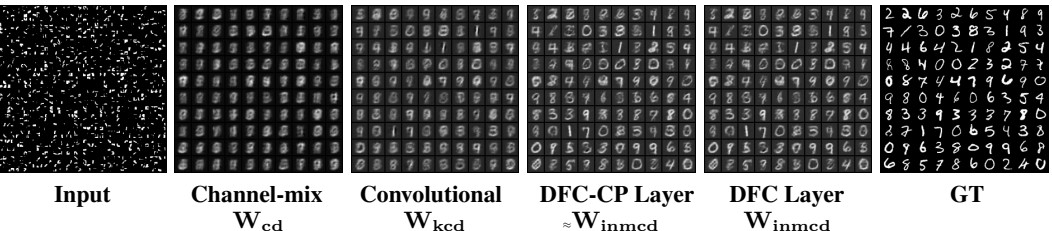

| **Input** | **Channel-mix** $\mathbf{W_{cd}}$ | **Convolutional** $\mathbf{W_{kcd}}$ | **DFC-CP Layer** $\approx \mathbf{W_{inmcd}}$ | **DFC Layer** $\mathbf{W_{inmcd}}$ | **GT** |
|---|---|---|---|---|---|

Figure 4: **Qualitative Comparison on MNIST dataset.** Augmenting the size of the weights tensor boosts output quality. Our DFC variant (DFC-CP) uses global receptive field and dynamic weights prediction and is able to generate clean outputs and sharp digits. Moreover, it uses CP decomposition to reduce complexity, performing on par with the computationally expensive DFC layer.

to leverage the entire content of the encoded features while adapting its weights depending on the input. Although DFC achieves the best performance, it is also the layer with the highest complexity. Notably, our CP-Decomposed DFC layer approximates DFC behavior at a fraction of the computational cost. The other layers, which are simplified cases of DFC, have lower complexity but also exhibit significantly lower performance.

**Classification, Segmentation, Detection** We extend the previous section by scaling up our analysis to different classes of architectures consisting of a stack of the compared layers, and three large-scale well-known CV applications: Image Classification on the ImageNet dataset, Object Detection, and Segmentation on the COCO dataset. In a similar spirit as before, we compare side-by-side four types of architectures created by progressively silencing various characteristics of the DFC layer. We use Former (i.e. a stack of DFC layers), MLP (i.e. a stack of FC layers), CNNs (i.e. a stack of Convolutions), and Linear Network (i.e. a stack of Channel-mixing Layers). To separate the contribution of macro design choices and building blocks, we fix an overall network design. Specifically, following the best practice of Liu et al. (2021b), we use a 4-stages hierarchical network with a stage compute ratio of 1:1:3:1. We build each stage as a stack of layers and GeLU non-linearities. We explore two different sizes: 15M parameters and 28M parameters. Given the well-known correlation between network complexity and final performance, we roughly match FLOPs and parameters count across methods with two strategies: i) we control for each method the number of overall features used at every stage, and ii) we use CP Decomposition to separate the tensors of weights into smaller matrices.[5] Table 1 reports the performance of Chop'D Former of equation 10 against its simplified variants: first, a CP-Decomposed MLP which drops the characteristic of dynamic weights but still achieves global reasoning via a spatial layer; then, two variants of CP-Decomposed CNNs (Lebedev et al., 2014; Howard et al., 2017) which drop global reasoning in favor of local processing. The first variant uses depth-wise convolutions as spatial layers, while the other uses non-adaptive average pooling to mix spatial information. Lastly, as a lower bound for performance, we use a CP-Decomposed linear network which is only able to process spatial information through the four pooling layers across stages. Results clearly show a performance progression that closely mimics the small-scale scenario and is consistent across tasks and architecture sizes. Remarkably, the linear network can generalize relatively well across tasks, even if its spatial receptive field is only 1 pixel

---

[5]We refer to supplementary materials for implementation details, and training hyperparameters.

| Layer | | Architecture | Complexity | | Classification | | | | Detection | | | Segmentation | | |
|---|---|---|---|---|---|---|---|---|---|---|---|---|---|---|
| Type | Weights | | P(M) | F(G) | T1 | T5 | v2 | Real | $AP^b$ | $AP^b_{50}$ | $AP^b_{75}$ | $AP^m$ | $AP^m_{50}$ | $AP^m_{75}$ |
| DFC-CP | $\approx \mathbf{W_{imncd}}$ | Former (Chop'D) | 15 | 2.4 | **80.9** | **95.4** | **69.6** | **86.6** | **40.1** | **61.4** | **43.8** | **37.1** | **58.6** | **39.6** |
| | | | 28 | 4.5 | **82.0** | **95.6** | **70.6** | **86.7** | **42.4** | **63.6** | **46.7** | **38.7** | **60.5** | **41.6** |
| FC -CP | $\approx \mathbf{W_{mncd}}$ | MLP | 15 | 2.4 | 78.5 | 93.2 | 66.5 | 85.0 | - | - | - | - | - | - |
| | | | 28 | 4.5 | 80.7 | 95.2 | 69.1 | 86.0 | - | - | - | - | - | - |
| Conv-CP | $\approx \mathbf{W_{kcd}}$ | CNN (Dw-Conv) | 15 | 2.4 | 78.9 | 94.4 | 67.6 | 85.2 | 38.7 | 60.1 | 41.9 | 35.8 | 57.0 | 38.2 |
| | | | 28 | 4.5 | 80.9 | 95.1 | 69.2 | 86.0 | 41.5 | 63.0 | 45.6 | 38.2 | 60.2 | 41.0 |
| Conv-CP | $\approx \mathbf{P_{kcd}}$ | CNN (Pool) | 15 | 2.4 | 78.5 | 94.0 | 67.0 | 84.8 | 38.0 | 59.5 | 41.3 | 35.5 | 56.6 | 37.6 |
| | | | 28 | 4.5 | 80.6 | 95.0 | 68.8 | 85.8 | 40.7 | 62.6 | 44.4 | 37.3 | 59.7 | 39.8 |
| Linear-CP | $\approx \mathbf{W_{cd}}$ | Linear | 15 | 2.4 | 73.9 | 91.4 | 60.9 | 80.8 | 29.7 | 50.4 | 31.0 | 28.9 | 47.7 | 30.4 |
| | | | 28 | 4.5 | 76.3 | 92.7 | 63.7 | 82.6 | 30.9 | 52.0 | 32.5 | 30.0 | 49.5 | 31.6 |

Table 1: **Comparisons among CP decomposition for different Class of Architectures** on Large scale classification on Imagenet and Detection and Segmentation on COCO using Mask-RCNN and a 1× training schedule. Chop'D Former approximates via CP decomposition DFC layers and outperforms less complex Neural Networks. Note that MLP cannot process the input of variable sizes and thus cannot be used as the backbone for Detection and Segmentation tasks.

| | Architecture | | | | Classification | | |
|---|---|---|---|---|---|---|---|
| Name | Type | Token-Mixer | Adaptivity $(i, n)$ | Receptive Field | Params (M) | Flops (G) | T1 |
| CoAtNet-0 (Dai et al., 2021) | Hybrid | Conv/MBConv/Global-SA | i,n | 3x3/Global | 25 | 4.2 | 81.6 |
| Poolformer-S36 (Yu et al., 2022) | CNN | Pooling | - | 3x3 | 31 | 5.0 | 81.4 |
| ConvNext-T (Liu et al., 2022) | CNN | Depthwise-Conv | - | 7x7 | 29 | 4.5 | **82.1** |
| RSB-ResNet-50 (Wightman et al., 2021) | CNN | Convolution | - | 3x3 | 26 | 4.1 | 79.8 |
| Swin-T (Liu et al., 2021b) | DCNN | Local-Self Attention (SA) | $i, n$ | 7x7 | 29 | 4.5 | 81.3 |
| GFNet-H-S Rao et al. (2021) | MLP | FFT | $n$ | Global | 32 | 4.6 | 81.5 |
| gMLP-S Liu et al. (2021a) | Former | Gated-MLP | $i, n$ | Global | 20 | 4.5 | 79.6 |
| Deit-S (Touvron et al., 2021) | Former | Global-SA | $i, n$ | Global | 22 | 4.6 | 79.8 |
| Chop'D Former - S | Former | Gated-SAT | $i, n$ | Global | 28 | 4.5 | 82.0 |

Table 2: **Comparisons with other architectures** for Large Scale Classification. Methods are trained on Imagenet-1K input image size 224 x 224 and have complexity between 4 and 5 GFLOPS.

wide ($\sim$ 74 T1, $\sim$ 30AP$^b$ and $\sim$ 29AP$^m$ for the smallest of the two sizes). As apparent from the table, the CNNs achieve better results, but the use of spatial information is still limited by local processing and shared response among spatial positions and instances. Interestingly, forcing a static global receptive field is not helpful, as shown by the fact that the MLP network does not significantly outperform the CNNs. Moreover, the MLP network cannot process inputs of various sizes and cannot be used as a backbone for detection and segmentation tasks. On the contrary, our Chop'D Former network approximates a set of DFC layers, calibrates its weights according to the input, and can integrate long-range interactions, outperforming CNNs variants by a large margin on both the 15M and 28M parameter variants. Chop'D Former gains an impressive $+2$ and $+1.1$ T1 in Classification. Similarly, Chop'D Former boosts results by $+1.4$ and $+0.9$AP$^b$ in detection and $+1.3$ and $+0.5$AP$^m$ in segmentation. Comparisons against state-of-the-art networks of comparable size are presented in Table 2 and expanded in the supplementary material. Without bells and whistles, Chop'D Former remains competitive against various architectural designs and offers a good trade-off between complexity and accuracy. It maintains a minimal gap with the best-performing method ($-0.1$ T1) and vastly outperforms other established Former architectural variants ($+2.2$ T1).

## 5  CONCLUSION

This work presents a new general layer for neural networks, "the DFC", a non-linear generalization for an FC layer, and a new architecture design, "the Former", built as a stack of DFC blocks. A DFC is dynamic, spatially adaptive, and fully connected but demands high computational requirements for deployment in real-case scenarios. To use Former architectures in CV applications, we propose to look through the lens of a unifying framework, based on CP Decomposition and Einstein notation, able to disentangle the individual characteristics of DFCs into separate components. Hence, we cast Transformers and their variants as CP-Decomposed Formers using different assumptions on the factor matrices and, consequently, distinct inductive biases. Then, we propose the Chop'D Former, a new hierarchical backbone for CV that approximates DFC blocks via CP Decomposition, leveraging the entire range of interactions via five sequential operations, including a spatial-mixing module with cost independent of the number of input positions. Lastly, we empirically demonstrate how each characteristic of DFC contributes to the overall performance, and we show that our CP-Decomposed (a.k.a Chop'D) Former can achieve state-of-the-art results on various CV benchmarks.

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
