# OpenReview forum: "Chopping Formers is what you need in Vision"
_ICLR.cc/2023/Conference — Submitted to ICLR 2023_

### Official Review · Reviewer_cu3r · 2022-10-24

**Confidence:** 5
**Correctness:** 2
**Technical Novelty And Significance:** 3
**Empirical Novelty And Significance:** 2
**Recommendation:** 3

**Clarity, Quality, Novelty And Reproducibility:**

The paper is well written and easy to follow. However, it is not clear that this new kind of architecture will be competitive with other general architectures such as transformers which are very efficient on different computer vision and NLP tasks and scale extremely well.The results seem reproducible.

**Strength And Weaknesses:**

Strengths:
- The subject treated in the paper is interesting as well as the different operations used in the decomposition to have dunamic layer on the different axis.
- The paper is quite well written and easy to follow.

Weakness:

- Hierarchical aspect: The decomposition of the dynamic block is interesting. However the hierarchical aspect of the architecture adds complexity and noise to the analysis. As we can see in Appendix Table 2 in order to design ChopForme we have to define the different pooling stage, the width and the depth of each. This complexify the architecture in comparison to vanilla transformers. It would be better to do an ablation of the architecture without the hierarchical aspect.

- Tasks:
The paper proposes an architecture presented as general but there are only tasks of computer vision. It would be interesting to complete it with NLP tasks to see if the architecture is competitive with transformers.

- Missing SOTA architecures: The architectures reported for comparison in the paper are not SOTA. In Table 1 appendix and Table 2 main paper. The following architecture should be added: EfficientNet-v2 [1], CoatNet [2], LeViT[3], MaxViT [4] The results of DeiT can be updated with those presented in the DeiT III paper [5].

[1] Tan et al., EfficientNetV2: Smaller Models and Faster Training
[2] Dai et al., CoAtNet: Marrying Convolution and Attention for All Data Sizes
[3] Graham et al., LeViT: a Vision Transformer in ConvNet's Clothing for Faster Inference
[4] Tu et al., MaxViT: Multi-Axis Vision Transformer
[5] Touvron et al., DeiT III: Revenge of the ViT

- Missing metrics: Several metrics are missing in the different tables to evaluate the tradeoffs of the proposed architecture as well as the ease of use. In addition to parameters and FLOPs, peak memory and speed should be reported.

- Only small architectures:  The paper considered only relatively small architectures. Ablations are done with the tiny version and the largest model reported is smaller than ViT-B. General architectures like transformers generally scale very well so it would be interesting to add results with larger architectures to see if the method scales. The comparison Table 2 in the main paper are only with small architectures.

- Overfitting evaluation: For the evaluation on ImageNet there is no measure of overfitting. But there is no separate validation set for ImageNet. It is necessary to add evaluation on ImageNet v2 in order to measure the level of overfitting of the proposed method. Especially since the more general an architecture is, the higher the risk of overfitting.

- Segmentation & Detection: There is no large comparison in segmentation and detection of the proposed method with other approaches like Swin or Deit III. It is important to make this kind of comparison in order to determine if the architecture is competitive on different tasks

**Summary Of The Paper:**

The purpose of the paper is to propose a more general architecture than transformers. The proposed architecture is adaptive according to the different instance axis, spatial dimension and channels. In order to limit the complexity of such an approach the paper adopts a decomposition of the different operations. The paper evaluates in image classification, segmentation and detection.

**Summary Of The Review:**

The idea of the paper is interesting but it lacks a lot of analysis to show that this approach is competitive with well established general architectures like transformers. Indeed, there is no experiment in the paper that shows the advantage of the proposed architecture over the transformers on different vision tasks like segmentation, detection and image classification. There is no experience that shows that the proposed architecture generalizes to other domains like NLP tasks and that the models scale well.

---

> ### Author Response · Authors · 2022-11-28
> **Main Reply**
>
> We thank the reviewer. We do believe that the subject of this paper is interesting! He appreciates the clarify of the exposure and the main topic but points out how our paper could benefit from a larger set of analyses. **Following reviewer's suggestions, we considerably extended the experimental results**. The experimental results in the main paper (Table 1) and in the supplementary (Tables 6, 8, 9, Fig.1) show how Chop'D Former achieves better results than TransFormers architectures (DeiT, ViT, PVT, gMLP) and dynamics CNN (dynamic DWNet-T, Swin-T).
>
> Moreover, **we would like to clarify our contributions**. As stated in the "summary of changes" reply in this work we provide three major claims (1) (2), and (3). While we appreciate the constructive feedback given about the experimental validation of the proposed architecture (claim (3)), we would like to remind that a non-negligible part of our contribution consists in claims (1) and (2) (i.e. (1) the proposed generalization of neural network layers and (2) the use of the Einstein notation and CP decomposition to provide a common and elegant framework in which comparing existing layers).
>
> **Regarding claim (1)**: The generalization of layers for neural networks with a new dynamic-fully connected layer (DFC) is novel. We point out that linear fully-connected layers can be generalized using DFC. Moreover, we show how DFC can generalize many layers that the computer-vision community is frequently using but cannot be represented as linear operators, most notably dynamic layers such as squeeze-and-excitation and self-attention. We wish to stress that our discussion on the dynamic components in DFC layers is not trivial and therefore is a key insight and novel contribution of our work.
>
> **Regarding claim (2)**: We also propose to use the Einstein notation to deal with tensor algebra in deep learning.
> As pointed out by another reviewer, a clean way to analyze different layers and architectures minimizes the mathematical obfuscation existing in related work, increases the significance of our work, and will bring value to the community. As such, we kindly ask the reviewer to consider this perspective while reading our work.
>
> **Regarding claim (3)**: The reviewer agrees that idea of the paper is interesting but it points out how the submission could benefit from additional analysis and results.
>
> We wish to stress that *the main idea of the experimental section of this paper is to investigate which aspects of spatial reasoning (namely size of the receptive field, dependence on the input, and spatial adaptivity) are relevant to process vision data*. In fact, vision benchmarks are ideal for an analysis of the importance of different aspects of spatial reasoning, requiring local or long-range interactions depending on the input and task at hand.
>
> *To do so, we design our experimental validation with fairness in mind, since we are not primarily interested in fine-tuning our setup to squeeze a few extra accuracy points but instead aim to compare different methodologies with a fixed training setup and architecture. As such, we would like to highlight how the significance of the work does not necessarily require state-of-the-art results, and that application on NLP tasks, while interesting, could be saved for follow-up work.*
>
> Nevertheless, we appreciate the detailed feedback and all the suggestions, and we did our best, given the computation resources at our disposal and the available time frame to strengthen the experimental section of our work within its computer vision scope.
> We provided extensive additional results on Classification and downstream tasks (detection and segmentation)

---

> ### Author Response · Authors · 2022-11-28
> **Regarding claim (3) Part 1**
>
> **Classification**
> * **Hierarchical architectures.** We thank the reviewer for the suggestion on experimenting with Hierarchical architecture!
> In the experimental section, we experimented with a hierarchical architecture composed of 4 stages. We made this macro-design choice with fairness in mind, since Convolutional Neural Networks, used as baselines in our comparisons, are limited to local processing and traditionally rely on hierarchical networks to augment their effective receptive field.
> Nevertheless, we agree that verifying whether or not our DFC block design can generalize to isotropic architectures is an interesting question and, following Rs' suggestion, we extended section 3.2 of the supplementary material with an ablation on the topic. As visible from these new results, **isotropic and hierarchical architectures replicate the same trend**, providing evidence that DFC layers are beneficial independently of the hierarchical or isotropic variant of the network in which are used.
>
> * **Additional references.** We extended Table 6 of supplementary with additional relevant references, considering as contemporaneous papers published (available in online proceedings) after May 28, 2022 (as per https://iclr.cc/Conferences/2023/ReviewerGuide).
> **We wish to highlight that the contributions of these works are orthogonal to our method.**
> In this work, we have proven that Chop'D Former can achieve a spatial reasoning that is global, spatially adaptive and dynamic, with a complexity comparable to a traditional convolution (a local, spatially shared, and fixed module) .
> [1,2,3,4] focus on hybrid-networks, which propose strategies to combine the strengths of different spatial modules in one architecture. [1,5] leverage carefully-crafted training and architectural setups, an investigation that is not in the scope of our work.
> In particular:
>      * [1] EfficientNet-v2 uses a combination of training-aware neural architecture search and scaling to best performance on Imagenet.
>      * [2] CoatNet studies how to combine effectively the strengths of convolution and self-attention.
>      * [3] LeViT combines a small convnet and a multi-stage transformer in one stack.
>      * [4] MaxViT combines efficient convolution, sparse attention and a new spatial-reasoning module.
>      * [5] DeiT III proves how training strategies alone can vastly impact networks' performance.
>
>    For these reasons, we consider the contributions of these works as orthogonal to our method. We leave the investigation of how our
>    method can be combined with other spatial-reasoning strategies and ad-hoc training procedures to future work.
>
> * **Scaling analysis**: We generally agree with the reviewer that large-scale models could behave differently from smaller-scale architectures and that this is indeed an interesting question to investigate.
> In our original submission, we provide an in-depth analysis with the computation resources at our disposal and tested models composed of a stack of 12, 24, and 36 blocks. We refrained from investigating larger width, since fitting training on a very-large-scale model (e.g. 88 M or 160 M parameters) on a single machine could require close to a month of training.
>  In this round of reviews, we **extended section 3.1 of the supplementary material with an analysis of Chop'D Former scaling**, to provide additional insight into our model's performance across various sizes.
> In particular:
>   *  In Table 6 of supplementary we have provided a comparison with models of all scales.
>   *  In sec. 3.1 Fig 1.b we extended the results presented in the main paper with two extra architecture sizes. *When tested in the same condition, our DFC block keeps a steady scaling behavior*, consistently outperforming other layers across 4 different network sizes.
>   *  In sec 3.1 Fig 1.a we contextualize the Chop'D Former performance against other dynamic networks (i.e. built as a stack of dynamic layers) independently of their training setting and macro-architectural choices. This new analysis highlights how *Chop'D Former design exhibits a solid trade-off between performance and size when compared to architectures using similar layers* .
>
> * **On additional metrics**:
>    * **Efficiency metrics.** We use flops and parameter counts as measures of efficiency. Contrary to peak memory and throughput on GPU, these **two metrics do not depend on implementation**, hardware choice, and software choice, therefore providing a fair comparison among methods. As a reminder, our method uses a custom CUDA kernel, which makes comparisons on throughput on GPU difficult against other methods, which use native pytorch/tensorflow implementation.
>   *  **Overfitting metrics** We integrated results with two measure of overfitting: the cleaned-up ReaL validation set and ImageNet-V2 . **These metrics show a trend in agreement with standard T1 validation**, showing that although general, Chop'D Former architecture does not tend to overfit.*

---

> ### Author Response · Authors · 2022-11-28
> **Regarding claim (3) part 2**
>
> **Downstream tasks**
>
> To showcase the versatility of our DFC backbone, we followed the evaluation setup of the CVPR 2022 oral paper [1] and extended our experimental section with a new set of experiments and comparisons on downstream computer-vision tasks.
> These new sets of results, as visible in the supplementary material in Table 7, Table 8, and Table 9, **show how Chop’D Former can function as a competitive backbone for dense prediction tasks in computer vision.**
>
> * **We extended section 4.2** of supplementary with new experiments and analyses on **detection and instance segmentation**. In more detail:
>
>   * **Table 7** of supplementary tests the capacity of Chop'D Former to work as a feature extractor for a single-stage detector (RetinaNet). This new set of results shows how under the same complexity and training strategy, Formers outperform other classes of architecture by a large margin, with a *trend that is generalized across detectors and architecture sizes*. We showcase the significance of our results by repeating the experiments with three separate random initialization and two network sizes. We reported the best performance by each of the evaluated methods (CNN (DW) CNN (Pool), MLP, Former (Chop'D) ). Note that the use of our DFC layers provides +1.5 AP and +0.7 AP, while the difference in performance across runs was no bigger than 0.2 AP, further showcasing the significance of our results.
>
>   * **Table 8** of supplementary compares Chop’D Former of various sizes against alternative backbones for CV detectors, by extending the experimental results of the PVT transformers of [1,2], papers that use the same training setups and augmentation strategies as ours. These new sets of results show how, *under the same complexity and training strategy, our Former variant is capable to work as a solid backbone* for computer-vision tasks, achieving good results across different sizes.
>
> * **We extended section 4.3** of supplementary with experiments on the new task of **semantic segmentation**, further validating the effectiveness of Chop'D Former to serve as a backbone for different downstream computer-vision tasks. Table 9 evaluates Chop’D Former of various sizes as the backbone for FNP comparing its results against alternative networks.  With this new set of experiments, we show how our DFC layers are able to *extract meaningful dense features from the input images, even when using a simple segmentation head that only minimally processes the features*.
>
> [1] Yu, Weihao, et al. "Metaformer is actually what you need for vision." Proceedings of the IEEE/CVF Conference on Computer Vision and Pattern Recognition. 2022.
> [2] Wang, Wenhai, et al. "Pyramid vision transformer: A versatile backbone for dense prediction without convolutions." Proceedings of the IEEE/CVF International Conference on Computer Vision. 2021

---

> ### Comment · Reviewer_cu3r · 2022-11-29
> **Response to the rebuttal**
>
> I thank the authors for their rebuttal, I have some additional questions and comments:
>
> - Hierarchical aspect:
>
> Thanks for the ablation.
>
> - Tasks: The rebuttal states:
>
> *As such, we would like to highlight how the significance of the work does not necessarily require state-of-the-art results, and that application on NLP tasks, while interesting, could be saved for follow-up work.*  I agree for claims 1) and 2) but claims 3) (*We connect our formulation to existing architectures by showing how Transformer and its variants
> can be seen as a stack of CP-Decomposed DFC operands for neural networks.*)  is a very strong claim so it's important to check this or revise this statement.
>
> - Missing SOTA:
>
>  *"For these reasons, we consider the contributions of these works as orthogonal to our method."*
>
>  I politely disagree, the claim of the paper are:
>
> a) *We connect our formulation to existing architectures by showing how Transformer and its variants can be seen as a stack of CP-Decomposed DFC operands for neural networks.*
>
> b) *We propose ChoP’D Former, a new variant of Former architecture, which is able to approximate the full DFC with a complexity comparable to a convolution with a small kernel, and is able to match, if not improve, SoTA performance on several benchmarks, including large scale classification, object detection, and instance segmentation.*
>
> If there is a claim of more general architecture and SOTA architecture it is important to have extensive comparison for instance with variants of transformers architectures. But I agree the main messages of the different papers are different.
>
> - Missing metrics:
>
>  peak memory and throughput are not perfect but this also the case for  flops and parameter. Please provide this metrics with a vanilla implementation this is usually the case for most of the competing methods and allows to have an estimation of the cost of the training and inference. Can you also provide training logs for the ablations and the main results?
>
> - Only small architectures:
>
> Please provide results with bigger architecture. Currently the paper state: "For example, Chop’D Former performance is comparable with a large dynamic DWNet (83.2 vs 82.8) which uses almost 4 times its amount of parameters (162 vs 42)"  this is not convincing because we can take as a counter example MaxViT-B [1] (120M parameters 84.95) or a vanilla ViT-B [2] 83.8 (86.9M parameters). So it seems that there is no evidence that the architecture proposes scale. But this is one of the main characteristics of transformers. If the proposed approach is more general it must also demonstrate this scaling property.
>
>
> [1] Tu et al., MaxViT: Multi-Axis Vision Transformer
> [2] Touvron et al., DeiT III: Revenge of the ViT
>
> - Overfitting evaluation:
>
> Thanks for completing results on ImageNet-v2 and ImageNet real.
>
> - Segmentation & Detection:
>
>  For semantic segmentation on ADE20K it's better to use UperNet as in Swin or DeiT-III. Indeed, this allows for a more extensive comparison with different state-of-the-art approaches. Currently the comparison does not allow for comparison with state of the art approaches such as Swin.

---

### Official Review · Reviewer_T6EK · 2022-10-24

**Confidence:** 4
**Correctness:** 3
**Technical Novelty And Significance:** 3
**Empirical Novelty And Significance:** 3
**Recommendation:** 6

**Clarity, Quality, Novelty And Reproducibility:**

The paper is clear. Main ideas seem to be novel. Most parts seem to be reproducible.

**Strength And Weaknesses:**

Strength:

(*) The paper proposes an interesting perspective on formulation of neural network layers

(*) The idea of forming the proposed generalized fully connected, and then apply CP decomposition is novel as far as I know, and interesting. The modifications proposed for the decomposed components to make it more practical, such as using SAT, or combining gating operators, are also contributing to the technical novelty of the paper.

(*) On the image classification task, the performance of the proposed model is comparable to SOTA.

Weaknesses:

(*) The paper proposes a novel layer and model that can be used for image classification, but it there is not enough discussion comparing it to recent ideas that go beyond ViT (e.g Swin-Transformer and ConvNextT models). It could be interesting to examine these models with your notation and perspective as well, since these achieve similar (or even better) efficiency and accuracy for the classification task.

(*) More experiments comparing the proposed model as a backbone for visual tasks might be useful for better evaluation against other models, such as experiments for detection and segmentation.



**Summary Of The Paper:**

The paper proposes a novel neural network layer architecture, which is derived by a tensor rank decomposition (CP decomposition) to a generalized version of a fully connected layer (namely, weights tensor of the FC that is created as a general function of the input). This new layer by design can be dynamic and spatially adaptive, it can create global receptive fields, and can mix channel information, therefore can accommodate the tensor operations for many of the existing known NN layers, such as self-attention or convolution layers.

While the full version of the proposed generalized fully connected layer is costly and has many parameters, its CP decomposion version has reduced computational complexity that makes it comparable with SOTA vision backbones, such as Swin-Transformer and ConvNextT models.  The authors conduct experiments on ImageNet-1K and show comparable top-1 accuracy as well.


**Summary Of The Review:**

Novel architecture based on interesting perspective on formulation of neural network layers.

---

> ### Author Response · Authors · 2022-11-28
> **Reply**
>
> We thank the reviewer for the suggestions! We do think that looking at other models from our perspective is indeed an interesting direction!
> * **Swin-Transformer**: A Swin block processes the input $X_{iknc}$ using two point-wise convolutions and one local self-attention with a receptive field of size $K=7$ x $7$. A Swin architecture uses a cycling window strategy to divide the image in non-overlapping patches of size $K$. Each Swin block processes each patch independently from all the others. Therefore, in a Swin block the spatial-output size is $N=7$x$7$, the spatial-input size is $K<<M$, and the number of instances is augmented with the number of patches generated $i \in [1, IHW/K]$. Starting from the Gelu-non linearity function $\sigma$, each block processes the input as in a traditional vision transformer: $Y\_{ind} =\sigma(((X\_{ikn\underline{c}\,} \, \textcolor{purple}{U^{4}\_{\underline{c}r}})\_{i\underline{k}nr}\textcolor{MidnightBlue}{U^{123}\_{i\underline{k}nr}})\_{in\underline{r}}\textcolor{purple}{U^{5}\_{d\underline{r}}} +  B\_{nd})$ . Each Swin block can be seen as an approximation of a **dynamic convolutional layer** (section 1.2 of the supplementary material):
> $Y_{ind} \approx \sigma(X_{i\underline{k}n\underline{c}}\, W^{'}_{i\underline{k}n \underline{c}d}) $
>
> * **ConvNext**: is a hierarchical architecture built as a stack of ConvNext blocks. This architecture uses a traditional unfolding strategy to divide the image in $N$ overlapping patches. Each block of the ConvNext architecture processes the input $X_{iknc}$ using two point-wise convolutions and one depthwise convolution with a receptive field of size $K=7$ x$7$  ($N>K$).
> Starting from the Gelu-non linearity function $\sigma$, each block processes the input as: $Y\_{ind} = \sigma(((X\_{ikn\underline{c}\,} \, \textcolor{purple}{U^{4}\_{\underline{c}r}})\_{i\underline{k}nr}\textcolor{MidnightBlue}{U^{2}\_{\underline{k}nr}})\_{in\underline{r}}\textcolor{purple}{U^{5}\_{d\underline{r}}} +  B\_{nd}) $.
> A ConvNext block can be seen as an approximation of a **convolutional layer** done via CP decomposition (Kruskal Convolution [1]): $Y\_{ind} \approx \sigma(X\_{i\underline{k}n\underline{c}}\, W^{*}\_{\underline{kc}d})$.
>
> For comparison, we report the five-dimensional tensor of the dynamic fully-connected layer DFC and its CP decomposition: $W\_{imncd}=U^{1}\_{i\,\underline{r}}U^{2}\_{m\,\underline{r}}U^{3}\_{n\,\underline{r}}U^{4}\_{c\,\underline{r}}U^{5}\_{d\,\underline{r}} + \epsilon\_{imncd}$.
>
> *Comparing the weights tensors $W\_{imncd}$ with $W^{'}\_{ikncd}$ and $W\_{kcd}$ highlights key individual properties of these layers*. Both the Swin block and ConvNext block have the limitation of local processing. ConvNext block is static (i.e. non-dynamic) and uses shared weights.
>
> *Nevertheless, analyzing building blocks for neural networks is just one of the factors that influence performance on any given task.* Other key factors are finding optimal architectural macro-design (e.g. how to combine building blocks one after the other to create an architectural design, the depth-width ratio to use, the unfolding strategy used to create patches) as well as fine-tuning training techniques (e.g. augmentation strategies, optimization used) in order to achieve good efficiency/accuracy trade-off in specific tasks.
>
> In this work, we provide a new perspective on the formulation of neural network layers and investigate which aspects of spatial reasoning (namely size of the receptive field, dependence on the input, and spatial adaptivity) are relevant to process vision data. To do so, we design our experimental validation with fairness in mind, to compare different layers in a controlled training setup with the same architecture.
>
> To provide further evidence of the benefit of using DFC layers *we extended experimental results to ablate two macro-design choices*: the size of the architecture and the hierarchical aspect of the architecture. As visible from these new results, isotropic and hierarchical architectures replicate the same trend (section 3.2 of the supplementary material) and the same can be said for four different network sizes (section 3.1 of the supplementary material), providing evidence that DFC layers are beneficial in various settings.
>
> To showcase the versatility of our DFC backbone, we followed the evaluation setup of the CVPR 2022 oral paper [2] and *extended experimental section with a new set of experiments and comparisons on downstream computer-vision tasks*. These new sets of results, as visible in the supplementary material in Table 7, Table 8, and Table 9, show how Chop’D Former can function as a competitive backbone for dense prediction tasks in computer vision.
>
> *In future work, we plan to extend our framework to cover other macro-design choices and training procedures*.
>
> [1] "Speeding-up convolutional neural networks using fine-tuned cp-decomposition." (2014)
> [2] "Metaformer is actually what you need for vision." (2022)

---

> > ### Comment · Reviewer_T6EK · 2022-12-08
> > **response**
> >
> > I thank the authors for their response. I think the addition of the swim transformer and ConvNext formulations contributes to their paper. I also understand the novelty concerns about the proposed approach. I am borderline, and will keep my original score.

---

### Official Review · Reviewer_P57x · 2022-10-25

**Confidence:** 4
**Correctness:** 4
**Technical Novelty And Significance:** 3
**Empirical Novelty And Significance:** 3
**Recommendation:** 6

**Clarity, Quality, Novelty And Reproducibility:**

- (Clarity) The paper is clearly written and easy to follow.
- (Quality) The experiments were well organized and clearly addressed the research questions brought throughout the paper.
- (Novelty) The novelty is limited in the sense that technically what the paper proposed is a low-rank approximation of existing architectures.
- (Reproducibility) Source code is not provided, but implementation details and the pseudo-code in the appendix provide reproducibility.

**Strength And Weaknesses:**

- (C1) It was a pleasure to read a paper trying to minimize mathematical obfuscation by employing concise einstein notation.
- (C2) I agree with the claim that the five-dimensional tensor W in the paper is general enough to represent all trending models of the community as described.
- (C3) A study for how low-rank r affects the performance of ChoP'D former is still needed. I even failed to find what r the authors used for the experiments in the main manuscript and the supplement material.
- (C4) The tensor W is only for a single-input layer. I am curious how the multiple-input layer, like cross-attention layers, could be formulated with Einstien-tensor notation. Plus, if possible, how can they be CP-decomposed (and its effectiveness too.) Maybe converting the original transformer architecture (with a transformer decoder) for machine translation would be a great starting point.
- (C5) Performances for major downstream tasks: COCO detection/segmentation and ImageNet classification are notably good. However, there is a report [1] that approximation methods tend to perform better on the small-parameter regime and worse on large-parameter models. Thus I want to see how Chop’D former works on at least a BERT-base scale (12 blocks).
- (C6) Typo: wrong double quote character on page 2.

[1] Tay, Yi, et al. "Scaling Laws vs Model Architectures: How does Inductive Bias Influence Scaling?." *arXiv preprint arXiv:2207.10551* (2022).

**Summary Of The Paper:**

The paper first delivers a general five-dimensional tensor operator that can generalize many batched single-input layers that our community is frequently using. Then, the authors decompose the tensor with CP-decomposition; using this low-rank approximation of the tensor, they propose an architecture Chop'D former. Another low-rank approximation of self-attention is used (SAT) to lower the complexity. The architecture's performance was on par or better on the benchmarks like COCO detection/segmentation and ImageNet classification for small parameter models (~30M)

**Summary Of The Review:**

It would be nice if some of my questions (see C3-C5) were answered. I want to give this paper a borderline accept recommendation because I believe the contributions that this simple and universal notation will bring to the community would outweigh the limited novelty.

---

> ### Author Response · Authors · 2022-11-28
> **Reply**
>
> We thank the reviewer for the interesting and insightful feedback! We address remarks one by one below:
> * **C3: How low-rank r affects the performance $R$**
>
> We thank the reviewer for pointing that out! We extended the supplementary with a clarification on the choices of $R, C$, and $D$ with an ablation on the topic (section 3.3 of the supplementary material). In our implementation, the DFC weight tensor $W\_{inmcd}$ has the same number of input and output channels $C = D$, and the rank of the CP decomposition ($R$) is assumed to be a quarter of the original dimensionality. To study the choice of how $R$ impacts performance, we ablate its value in a new set of experiments. Table 3 reports performance and shows how, when $R$ decreases, the performance degrades since the decomposition is not able to approximate properly the weight tensor, and thus converges to a suboptimal solution.
>  * **C4: On multiple-input layer**
>
> Again, we thank the reviewer for the very interesting suggestion! We think that we could tackle this idea from two perspectives:
> * 1) A DFC layer (Eq. 3 main paper) could directly tackle the case of two streams of information by considering them as the same instance $i$ with two different sets of features stacked along the input channel dimension. In this case, the cross-attention layer would take in input a single "stacked" tensor $X_{imc}$ where the number of channels $c \in [1, 2C]$ is doubled. The cross-attention layer could use the same formulation of a standard self-attention (Eq 7 in the main paper) with a modification on the generation of weights $U^{123}_{imnr}$. In fact, It would only require two extra "masking" matrices, in charge of selecting a subset of channels for $Q$ and a different subset of channels for $K$.
> * 2) Einstein notation could be used to deal with two streams of information explicitly without using any concatenation. For example, it can be used to describe the bilinear layer. In this case, one can consider two streams of information as the tensors $X\_{mc}$ and $Z\_{md}$, of $M$ tokens and $C$, $D$ features. A bilinear layer parametrized by $W\_{cde}$ parameters creates the output as $Y\_{me} = X\_{m\underline{c}} W\_{\underline{cd}e} Z\_{m\underline{d}}$.
>  We leave the extension of our framework for future work, where we plan to provide an in-depth analysis of multiple-stream of information via Einstein notation and CP decomposition.
>
>
> * **C5: On Scaling**
>
> We generally agree with the reviewer that large-scale models could behave differently from smaller-scale architectures and that this is indeed an interesting question to investigate. In our original submission, we provide an in-depth analysis with the computation resources at our disposal and tested models composed of a stack of 12, 24, and 36 blocks. In this round of reviews, we refrained from investigating larger widths, since fitting training on a very-large-scale model (e.g. 88 M or 160 M parameters) on a single machine could require close to a month of training, yet we would like to address the concern of the reviewer and so we extended section 3.1 of the supplementary material with an analysis of Chop'D Former scaling up to 40M parameters, to provide additional insight into our model's performance across various sizes. 1) We extended the results in Table 1 of the main paper with two extra sizes. When tested in the same condition, our DFC block keeps a steady scaling behavior, consistently outperforming other layers across 4 different network sizes (Fig 1 (b)). 2) We contextualize the Chop'D Former performance against other dynamic networks (built as a stack of dynamic layers) independently of their training setting and macro-architectural choices. This new analysis highlights how Chop'D Former design exhibits a solid trade-off between performance and size when compared to architectures using similar layers (Fig 1 (a)). 3) We have provided a comparison with models of all scales (Table 6 of supplementary). In conclusion, we hope that these new sets of results can confirm the advantage of using our dynamic, spatially adaptive, and global reasoning layer when building neural network architectures of various sizes.

---

> > ### Comment · Reviewer_P57x · 2022-12-07
> > **Reply**
> >
> > I thank the authors for the number of their experiments done during the rebuttal period.
> > It answered my questions thoroughly.
> >
> > Though, as other reviewers pointed out, I still think the proposed model has limited novelty, which makes me hard to raise my recommendation from 6 to 8.

---

### Official Review · Reviewer_a5bE · 2022-10-25

**Confidence:** 4
**Correctness:** 2
**Technical Novelty And Significance:** 1
**Empirical Novelty And Significance:** 2
**Recommendation:** 3

**Clarity, Quality, Novelty And Reproducibility:**

Clarity: Poor.

Quality: Poor.

Novelty: Medium.

Reproducibility: N/A.

**Strength And Weaknesses:**

Pros:

- Have a try to unify nowadays models.

Cons:

This paper seems to be organized by (1)firstly unifying various networks; (2) then denoting networks with a CP format; (3) lastly designing ChoP’D to achieve a good performance on benchmarks. However, I have some concerns:
  - The unifying strategy is not strict and not novel. The paper claims to unify convolutions, fully-connected layers, and transformers into a category "Former".
    - The Transformers are not suitable for this category since they have non-linear structures (namely self-attention that has softmax function). In addition, this paper claims "Without lack of generality, we omit at this stage the Layernorm (LN) applied before every block and the residual connections". As the softmax and LN functions are both important components in a network, omitting them is not suitable. Therefore, it is unreasonable using such loose conditions to group Transformer and CNNs into one category.
    - For the CNNs and fully-connected layers, [1] has already unified them. And [2] also gives tensorial CNNs a unified representation to initialize them in one scheme. Therefore, the idea of unifying CNNs is not novel.
  - Unifying models into a CP format is already proposed by [3]. Equation (5) in this paper is the same as Equation (2) in [3]. Thus, such CP representation is also not novel.
  - For the modification of the Former to construct ChoP’D Formers, there seems some limited novelty.

- Unreadable writing:
  - This paper uses improper uppercase word formats like "Dynamic and Spatially Adaptive". Simply using them in lowercase is ok. For example, "Dynamic and Spatially Adaptive" -> "dynamic and spatially adaptive";
  - Grammar errors like "Vision Transformers (ViT) success has long ..." -> "The success of Vision Transformers (ViT) has long...";
  - Overlong paragraphs:
    - "Puzzle Reconstruction" on Page 7;
    - "Classification, Segmentation, Detection" on Page 8.
  - Inconsistent descriptions like "fully-connected" and "Fully Connected";
  - The quotation mark in Latex is written as `` '';
  - The citation should use citep and citet.

- For the performance, the proposed ChoP’D Formers seem only derives some comparable results to baselines, which are not significant. So, what are the remarkable advantages of ChoP’D?


Overall, regarding the above points, I vote for "rejection".

[1] Kohei, Hayashi, et al. "Exploring unexplored tensor network decompositions for convolutional neural networks." Advances in Neural Information Processing Systems 32 (2019).

[2] Yu, Pan, et al. "A Unified Weight Initialization Paradigm for Tensorial Convolutional Neural Networks." International Conference on Machine Learning. PMLR, 2022.

[3] Jean, Kossaifi, et al. "Factorized higher-order cnns with an application to spatio-temporal emotion estimation." Proceedings of the IEEE/CVF Conference on Computer Vision and Pattern Recognition. 2020.

**Summary Of The Paper:**

The paper proposed ChoP’D Formers based on CP decomposition representation. Some modification is implemented on the proposed "Former". And experiments show some comparable results.

**Summary Of The Review:**

The paper proposed a newly designed model based on CP decomposition. However, I still have some concerns as mentioned above. Therefore, I tend to give "rejection".

---

> ### Author Response · Authors · 2022-11-28
> **Part 1**
>
> We thank the reviewer for the chance to clarify our contributions, and we appreciate the constructive feedback given. We respond to each specific concerns below.
>
> * **C1: "The paper claims to unify convolutions, fully-connected layers, and transformers into a category Former"**
>
> We understand the confusion! We try to clarify our contribution below.The main point to clarify is that our framework does *not* aim to unify blocks (conv, linear layers) with architectures (CNNs, transformers). We first provide a common formulation to define basic building blocks and then use this common formulation to define different types of architectures.Specifically, the core claim of this paper is to present a general building block for neural networks, able to generalize and unify many layers that the computer-vision community is frequently using. We highlight that many of these layers cannot be represented as linear operators (e.g. squeeze-and-excitation and self-attention) and as such cannot be framed as special cases of fully-connected layer. As described in section 2.2. on the main paper, let us recall that a **linear layer** is characterized by having the weights tensor $W$ independent of the input and therefore parametrized with a **set of learnable parameters** that is static with respect to the input. On the other hand, a **dynamic layer** is characterized by a **weights tensor that is a function of the input, i.e. $W=g(X)$, resulting in a layer $Y(X) = X g(X)$ that is a non-linear function of $X$**. Examples of *static* blocks are convolution and fully connected layers. Examples of *dynamic* layers are self-attention and squeeze-and-excitation layers.This link between non-linear and dynamic operators leads to the definition of our general DFC layer. **We wish to stress that the ability to frame dynamic, non-linear, blocks are a unique feature and key insight of our work.** In fact, building on top of our DFC layer, we can define instances of different archetypal neural networks. Specifically, our formalization can differentiate between static neural networks (stack of static linear layers), and dynamic neural networks (stack of dynamic non-linear layers). Examples of static neural networks are CNNs and MLPs. Examples of dynamic neural networks are Formers and Transformers.
>
> * **C2: "[1] has already unified CNNs and fully connected layers. [2] gives tensorial CNNs a unified representation."**
>
> In this work, we do not unify CNNs and fully-connected layers. By design, convolutional layers can be framed as special cases of fully-connected layers as discussed in [5,6].
> In this work, we explicitly tackle *dynamic* networks, and in particular networks built as a stack of dynamic fully connected layers (Formers). Related work [1,2] focuses on following up to [4] and exploring alternative tensor decompositions to factorize *static* Convolutional layers into lower-rank tensors. In particular: [1] extends [4] by characterizing a set of alternative decompositions for CNNs via hypergraphical notation. [2] discusses initialization methods for Tensorial Convolutional Neural Networks via hypergraphical notation. **There are two key differences** between the contributions of our paper and those of [1] and [2].
> First, **both papers focus on tensor decomposition for Convolutional layers, which are local and not dynamic.** Here, we focus on DFC layers.
> The other key difference between us and these methods is in the notation. [1] and [2] use **hypergraphical notation structures to discuss decompositions**. Here, we propose to use Einstein notation to deal with tensor algebra. Compared to the use of the Einstein notation, this graphical representation lacks simplicity and **still requires tensor algebra to be discussed in practice in the text.**
>
> [1] Hayashi, Kohei, et al. "Exploring unexplored tensor network decompositions for convolutional neural networks." Advances in Neural Information Processing Systems 32 (2019).
> [2] Pan, Yu, et al. "A Unified Weight Initialization Paradigm for Tensorial Convolutional Neural Networks." International Conference on Machine Learning. PMLR, 2022.
> [3] Kossaifi, Jean, et al. "Factorized higher-order cnns with an application to spatio-temporal emotion estimation." Proceedings of the IEEE/CVF Conference on Computer Vision and Pattern Recognition. 2020.
> [4] Lebedev, Vadim, et al. "Speeding-up convolutional neural networks using fine-tuned cp-decomposition." arXiv preprint arXiv:1412.6553 (2014).
> [5] LeCun, Yann. "Generalization and network design strategies." Connectionism in perspective 19.143-155 (1989): 18.
> [6] Goodfellow, Ian, Yoshua Bengio, and Aaron Courville. Deep learning. MIT press, 2016. (Chapter 9)

---

> > ### Author Response · Authors · 2022-11-28
> > **Part 2**
> >
> > * **C3: " The Transformers are not suitable for this category since they have non-linear structures (softmax) and Layernorm (LN) "**
> >
> > Similarly to how we could define Convolutional Neural Networks as an architecture built as a stack of convolutional layers (with or without CP decomposition), In Eq. 4 of the main paper we define as a "Former" Network an architecture built as a stack of dynamic fully-connected layers (DFC). Looking at Eq. 7, one can see that, even if Transformers have non-linearities associated with their architecture (i.e. a GeLU function and a softmax function), this is completely in line with a Former architecture, built as a stack of *non-linear* extension of the FC layer (DFC).In particular, as described in section 2.3 paragraph "Transformer" and the footnote on page 5 in the main paper, **the softmax non-linearity function is used solely to define the function $g$ used to generate factor matrix of the CP decomposed DFC and is, therefore, perfectly compatible with our formulation.** Under the reviewer's suggestion, we add the softmax to our equations in the main paper and supplementary, but we wish to stress to **Eq. 7 is agnostic to the type of function used to create $U^{123}_{imnr}$**.
> > A similar discussion can be done for the normalization layers, which only impact the assumptions on the factor matrices of the CP decomposition. Specifically, from the perspective of the learnable weights in the CP decomposition, we can consider Layer Norm as a simple affine transformation of the input, after standardization (which is a fixed, non-learnable, operation) that can be implemented with a simple convolution. Therefore the use of Layer Norm does not impact any conclusion in our formulation.
> >
> > * **C4: Eq. (5) in this paper is the same as Eq. (2) in [3]**
> >
> > **Eq. (5) in this paper is *not* the same as E. (2) in [3]**.
> >
> > Eq. 5 of the main paper defines the novel CP decomposition for the general five-dimensional tensor operator of the dynamic fully-connected layer DFC:
> > $W\_{imncd}=U^{1}\_{i\underline{r}}U^{2}\_{m\underline{r}}U^{3}\_{n\underline{r}}U^{4}\_{c\underline{r}}U^{5}\_{d\underline{r}}+ \epsilon\_{imncd}$
> >
> > Eq. 2 of [3] (originally Eq.5 of [4] ) defines the CP decomposition for the three-dimensional operator of a convolution (Kruskal convolution):
> > $W^{*}\_{kcd}=U^{2}\_{k\underline{r}}U^{4}\_{c\underline{r}}U^{5}\_{d\underline{r}} + \epsilon_{kcd}$.
> >
> > In the above equations, $i$ indexes images in a batch. $n$ and $m$ index input and output spatial positions. $c,d$ index input and output channels. **Note that height and width of the image are condensed over one dimension** and as such their range is the entire size height*width of the image (i.e. $HW$). Therefore, in this notation, the Kruskal convolution is associated with a 3D tensor, with $k$ indexing the total size of the receptive field. For example a receptive field $3$x$3$ would appear as a $k \in [1,9]$).
> >
> > **Comparing the weights tensors $W\_{imncd}$ with $W^{*}\_{kcd}$ highlights differences** :
> > 1) Kruskal convolution has a **local** receptive field, a DFC has a global receptive field ($m\in [1, HW]$, $k \in [1, K]$ with $K<<HW$);
> > 2) Kruskal convolution has **shared** weights, DFC is spatially adaptive. DFC has different weights for every output spatial position $n \in [1, HW]$, Kruskal convolution shares the same set of weights for every output spatial position;
> > 3) DFC is **dynamic**, while Kruskal convolution shares the same set of weights for every image in the batch. This is reflected in the additional dimension of the DFC weights tensor: DFC has different weights for every image $i \in [1, I]$ in the batch.
> >
> > * **C5: "the proposed ChoP’D Formers seem only derives some comparable results to baselines"**
> >
> > The main idea of the experimental section of this paper is to investigate which aspect of spatial reasoning (i.e. the size of the receptive field, dependence on the input, and spatial adaptivity) is relevant to process vision data. As such, we would like to stress that the significance of the work is not necessarily confined to producing state-of-the-art results in specific benchmarks (against highly specialized methods specifically fine-tuned only for this). **Our main goal is to show that, armed with a clear understanding of fundamental characteristics of basic building blocks for neural networks, we can provide competitive results without any hyperparameter search and architecture design ablations.** To provide further evidence that Chop'D Former can act as a solid backbone for computer-vision applications, we considerably extended supplementary material to validate our claims. In supplementary, we show that Chop'D Former reaches good performance when compared to methods built as a stack of dynamic layers like ours, but trained using different setups and macro-design (sec 3.1 Fig 1. a), and keeps a steady advantage when compared to methods built as a stack of non-dynamic layers (sec 3.1 Fig 1)

---

> > > ### Comment · Reviewer_a5bE · 2022-12-04
> > > **Reply**
> > >
> > > Thanks for the response from the authors. However, based on the knowledge of [1][2][3], I do not feel especially much novelty in the proposed "Formers" in CP representation. In addition, the formulation without considering non-linear conditions (namely loosen condition) makes a limited unification, i.e., linear parts of the mentioned models. Moreover, I prefer to see a significant improvement based on the proposed method. However, Table 2 in the main paper only shows a weak performance. In Figure 1(a) of the appendix, Chop’D Former shows good performance when parameters < 42M. However, most baselines show the performance of a size of around 80M, which is also an important size as almost all baselines in Table 6 have reported such a magnitude. By the way, it is also weird for Swin to show a sharp decrease in around 60M size. Based on the above consideration,  I would like to maintain the score.

---

### Official Review · Reviewer_8HGv · 2022-10-25

**Confidence:** 4
**Correctness:** 3
**Technical Novelty And Significance:** 3
**Empirical Novelty And Significance:** 2
**Recommendation:** 5

**Clarity, Quality, Novelty And Reproducibility:**

- the paper is well written and is easy to follow
- the proposed architecture is simple and should be easy to implement and reproduce the experiments.

**Strength And Weaknesses:**

- the proposed generalization of a linear layer provides a simple novel view on convolutional and Transformer networks

Experimental validation is lacking.
- while the authors propose a generalization of linear layers and Transformers, the evaluation is done only on computer vision benchmarks. Applying the proposed approach to non-computer vision tasks would provide stronger evidence for the claims
- in the puzzle reconstruction experiment, it is not clear what information is provided by the Convolutional, Spatial mix, Pooling and Channel mix curves. As the task requires global receptive field, it is expected that the layers without it would be able to the reconstruction task. A convolutional network with more layers should be able to handle the task as well. A comparison with linear layer only should be sufficient to validate the claims.
- in object detection, 1x schedule is not sufficient for experimental validation, as some networks might need more iterations to reach peak performance.
- citations [1] and [2] are missing. These can also serve as stronger baselines for the experiments.
- it is not clear is the results in Table 1 are statistically significant, or are within std of training with different random seed.

[1] Wang et al., Non-local Neural Networks
[2] Hu et al., Squeeze-and-Excitation Networks

**Summary Of The Paper:**

The authors attempt to generalize linear layers with layers in which weights are produced by a function on inputs, motivated by the success of Transformers. Motivated by the CP decomposition, they extend the proposed layer to be more computationally efficient, and suggest a variant of a convolutional neural network architecture. They validate that the proposed architecture works on par with common approaches in image classification and object detection benchmarks.

**Summary Of The Review:**

While the attempt to generalize linear layers and connect with Transformers is valuable, it seems like the resulting architecture is simply another variation of convolutional neural networks. A stronger and more thorough experimental validation is needed to validate the claims and novelty.

---

> ### Author Response · Authors · 2022-11-28
> **Reply**
>
> We thank the reviewer for the insightful feedback! He appreciates the value of our main claims and the clarity of our exposure. Nevertheless, he points out how the paper would benefit from extra experimental validation.
>
> The main idea of the experimental section of this paper is to investigate which aspects of spatial reasoning (namely size of the receptive field, dependence on the input, and spatial adaptivity) are relevant when designing neural network architectures. As a starting point, we chose computer-vision tasks for our analysis since they offer a good benchmark to evaluate the importance of different aspects of spatial reasoning, but we plan to expand our results further in future works. In this paper, however, **we took on board the suggestion of the reviewer and extended our experimental section within the computer vision scope**, and as a result in the revised manuscript, to the best of our ability in the given timeframe and with computational resources at our disposal, we have substantially extended the experimental sections with a large number of additional experiments. **We hope that this will clarify the ability of our method to perform as a solid computer vision backbone across more tasks and sizes**. In more detail:
>
> **Puzzle Reconstruction.**
> We have shortened the paragraph to ease reading.
> The main idea of the experimental section is to introduce various aspects of spatial reasoning. As such, we use this simple setting as a chance to discuss different variants of DFC and their complexity, as shown in Figure 3 (right). In particular, we wanted to show how a DFC approximation (ours) has complexity comparable to a Convolutional layer but performs as well as a standard DFC. We wanted to point out that a Convolutional layer could come in different flavors of complexity (Convolutional, Spatial mix, Pooling).
>
> **Classification.**
> We expanded supplementary with a new set of ablations and analyses (sections 3.1, 3.2, and 3.3), showing how our DFC layer benefits architectures in a range of controlled setups and sizes.
>
> **Downstream Tasks.**
> To showcase the versatility of our DFC backbone, we followed the evaluation setup of the CVPR 2022 oral paper [1] and extended our manuscript with a new set of experiments and comparisons on downstream computer-vision tasks, as visible in additional material in Table 7, Table 8, and Table 9. Our goal, we recall, is to keep the comparison among methods fair. So we used well-established training and augmentation strategies in all our benchmarks and we are working on extending our findings to cover larger training setups. For Table 7, we repeated the experiments with three separate random initialization and two network sizes. Note that the use of our DFC layers provides +1.5 AP and +0.7 AP, while the difference in performance across runs was no bigger than 0.2 AP, further showcasing the significance of our results.
>
> We hope that all these new results plus the ones in the original submission will convince the reviewer that our analysis of building blocks for neural networks via Einstein notation and CP decomposition is general (claims 1 and 2).
>
> [1] Yu, Weihao, et al. "Metaformer is actually what you need for vision." Proceedings of the IEEE/CVF Conference on Computer Vision and Pattern Recognition. 2022.

---

### Author Response · Authors · 2022-11-28
**Summary of Changes**

Dear area-chair and reviewers, please find here a response and updated manuscript. We are grateful to all reviewers for their insightful feedback. We have carefully analyzed all the reviews, responded to each comment, and updated content and experimental analysis to address remarks. In the past few weeks, we have spent considerable amount of time to run as many additional experiments as possible, as evidenced by the revised version of the manuscript, and based on the new results we can now provide our responses. We hope to have a fruitful exchange with all reviewers in the remainder of this discussion stage.

The summary of the claims of our paper is as follows:
* (1) We formalize a general layer for neural networks: a dynamic fully connected layer (i.e. DFC). DFC is a *non-linear* generalization of a fully-connected layer, where the weights are created *dynamically* as a function of the input.
* (2) We provide a simple and novel framework based on Einstein notation and CP decomposition to describe existing layers for neural networks.
* (3) We showcase our framework by formulating a new and general CP decomposition for DFC, targeting computer-vision applications. We provide evidence on how, despite performance in challenging benchmark for CV is driven by a mix of causes (i.e. layers used, architectural choices and training setup), reasoning about layers structure is enough to achieve a good trade-off between efficiency and performance.

The summary of the main changes for our supplementary material is as follows:

* We considerably extended the experimental results, providing an in-depth evaluation against other alternative backbones for computer-vision tasks to further validating our claims. In particular:
     * We **extended section 3.1 of the supplementary** material with an overview of the **trade-off between performance and size**. With this new set of experiments and analysis we showcase how Chop’D Former stacks favorably against similar methods and provide additional evidence in support of claims (1) and (3).
    * We **extended section 3.2. of the supplementary** material with a new ablation on **isotropic architectures**. With this new set of experiments, we showcase that our DFC block design can generalize to models without pooling, providing additional evidence in support of claims (1) and (3).
    * We **extended section 3.3 of the supplementary** material with ablation on how the **choice of the rank $R$** affects the performance of ChoP’D Former, clarifying our choice for tensors dimensionality $R$, $C$, $D$.

    * We **extended section 4.2 and 4.3 of the supplementary** material with additional results on the **downstream task of detection**. With this new set of results, we showcase how Chop'D Former design can serve as a solid backbone for computer-vision tasks, even without searching for ultra-specialized architectures or fine-tuned training settings.  In particular:
        * We extended section 4.2 of supplementary with results on an **additional detector**, to provide evidence that our design can generalize across architectures for downstream tasks.
        *  We extended section 4.2 of supplementary with **comparisons against alternative CNN and ViTS backbones**, to provide evidence that our design can achieve a good trade-off between performance and size.
        *   We extended section 4.3 of the supplementary material with results for the task of **semantic segmentation**, providing evidence that our design can extract meaningful dense features for different downstream tasks.

The summary of the changes for our main manuscript is as follows:
 *  We amended the main paper as suggested by reviewers. In particular:
     * We added relevant citations.
     * We extended the related work of the "Tensor Decomposition for Neural Networks" paragraph of the main paper to discuss relevant literature.
     * We extended the results to include **two additional measures of overfitting**. With these additional measures, we show the capacity of Chop'D Former to generalize, providing further evidence in support of claims (1) and (3).
    * We also improved the readability of the text and corrected the minor typos that were present in the original submission.

Overall, these changes strengthen and improve our submission. We address the reviewer's remarks individually below.

---

### Decision · Program_Chairs · 2023-01-20

**Decision:**

Reject

**Justification For Why Not Higher Score:**

Weak baselines. Minimal empirical gains.

**Justification For Why Not Lower Score:**

n/a

**Metareview: Summary, Strengths And Weaknesses:**

In this work, the authors propose a new neural network architecture based on a tensor-rank decomposition (CP decomposition) which can generalize convolution and self-attention layers. In particular, the authors test the efficacy of their method on puzzle reconstruction with MNIST, detection and segmentation on COCO, and image classification on ImageNet. The proposed layer achieves competitive and at times favorable performance given a computational budget across an array of these vision tasks.

The reviewers commented positively on the novel manner for unifying convolutional and Transformer layers, the application of the CP decomposition, and competitive numbers on ImageNet image classification. The reviewers also expressed concerns about (1) the overall novelty of the resulting architecture, (2)  a lack of experiments on non-computer vision tasks, (3) relationship with previous works on unifying CNN’s and Transformers, and (4) lack of strong improvement over previous methods. During subsequent discussion, the primary issues about (3) and (4) were not resolved and no reviewer raised their score to champion acceptance of the paper.

Although I sympathize with the desire to only evaluate on vision tasks to make the problem more tractable for a conference-length submission, I am concerned that (3) and (4) specifically. To address (4) the authors do need to provide additional clarification in the manuscript about the relationship with prior work attempting to generalize across layers and architectures. With regard to (3), one item of discussion focused on appropriate comparisons and which works were contemporaneous with the present work. Given a submission deadline of May 28, 2022, several of the papers highlighted – e.g. CoatNet (NeurIPS 2021), EfficientNetv2 (ICML 2021); MaxViT (ECCV 2022) – were all accepted prior to this cutoff deadline and should be considered valid prior work when arguing whether this model achieves SOTA performance. Given that SOTA performance is one of the central contributions in the Introduction, there should at the minimum be additional clarity from the authors about what aspects are being considered SOTA with respect to what baselines. Because of all of these unresolved issues, we can not accept this paper to this conference. I would instead encourage the authors to revise their manuscript and their claims accordingly and resubmit to a future venue.